# Simple Synthesis of Cobalt Carbonate Hydroxide Hydrate and Reduced Graphene Oxide Hybrid Structure for High-Performance Room Temperature NH_3_ Sensor

**DOI:** 10.3390/s19030615

**Published:** 2019-02-01

**Authors:** Chang Wang, Huan Wang, Dan Zhao, Xianqi Wei, Xin Li, Weihua Liu, Hongzhong Liu

**Affiliations:** 1Department of Microelectronics, School of Electronics and Information Engineering, Xi’an Jiaotong University, Xi’an 710049, China; wangc254@163.com (C.W.); wanghuan19960511@stu.xjtu.edu.cn (H.W.); zhaodan_xjtu@163.com (D.Z.); wei.wxq@163.com (X.W.); lwhua@mail.xjtu.edu.cn (W.L.); 2State Key Laboratory for Manufacturing Systems Engineering, Xi’an Jiaotong University, Xi’an 710049, China; hzliu@mail.xjtu.edu.cn; 3Guangdong Shunde Xi’an Jiaotong University Academy, NO.3 Deshengdong Road, Daliang, Shunde District, Foshan 528300, China; 4Research Institute of Xi’an Jiaotong University, No. 328 Wenming Road, Xiaoshan District, Hangzhou 311215, China

**Keywords:** gas sensor, cobalt carbonate hydroxide hydrate, reduced graphene oxide, room temperature, ammonia

## Abstract

A novel hybrid structure sensor based on cobalt carbonate hydroxide hydrate (CCHH) and reduced graphene oxide (RGO) was designed for room temperature NH_3_ detection. This hybrid structure consisted of CCHH and RGO (synthesized by a one-step hydrothermal method), in which RGO uniformly dispersed in CCHH, being used as the gas sensing film. The resistivity of the hybrid structure was highly sensitive to the changes on NH_3_ concentration. CCHH in the hybrid structure was the sensing material and RGO was the conductive channel material. The hybrid structure could improve signal-to-noise ratio (SNR) and the sensitivity by obtaining the optimal mass proportion of RGO, since the proportion of RGO was directly related to sensitivity. The gas sensor with 0.4 wt% RGO showed the highest gas sensing response reach to 9% to 1 ppm NH_3_. Compared to a conventional gas sensor, the proposed sensor not only showed high gas sensing response at room temperature but also was easy to achieve large-scale production due to the good stability and simple synthesis process.

## 1. Introduction

As an irritating odor and highly toxic gas, ammonia is widely found in industrial exhaust emissions, which is gradually threatening the health of humans and animals. Therefore, it is very crucial to develop highly sensitivity ammonia sensors and detection equipment.

Semiconductor gas sensors, all using metal oxide semiconductors as sensing materials, are widely applied to detect various gases owing to the good repeatability and low cost [1,2,3]. Xu et al. have adjusted the aspect ratio of In_2_O_3_ nanowires to prepare gas sensors with a great gas sensing properties to NO_2_ [4]. The SnO_2_ nanowire had good gas sensing properties for hydrogen [5]. Wagh et al. used modified ZnO thick film to prepare a highly sensitivity ammonia sensor [6]. In order to further improve the gas sensing properties, there were two research ideas in the academic circles all over the world. The first one was to increase the specific surface area of sensing materials by preparing porous materials, which could improve the gas sensing properties by providing more gas adsorption sites on the sensing materials surface [7,8]. The other was to prepare composite film gas sensors based on the excellent conductivity of graphene and the gas sensitivity of metal oxides [9,10]. Considering the high mobility and conductivity of graphene, the latter method has attracted increasing attentions from the gas sensors field.

In recent years, many works on gas sensors based on composite structures of graphene and sensing materials have been published [11,12,13,14,15,16,17]. RGO decorated TiO_2_ microspheres was fabricated by a hydrothermal method and showed excellent sensitivity and good selectivity to ammonia gas [16]. Feng et al. prepared an evenly dispersed RGO-SnO_2_ composite sensor by mixing the RGO obtained from thermal reduction with the commercial SnO_2_ nanoparticles. Although the sensor had good gas sensing properties at room temperature, its noise was very high [17]. Liu et al. has grown ZnO nanowalls on the RGO surface to realize a gas sensor with composite structure, which has realized the detection of nitrogen dioxide at room temperature [18].

Combining with the previous research, a new research idea has been gradually produced recently, which is to synthesize the ideas proposed above so as to further improve the gas sensing properties. ZnO nanoparticles loaded onto 3D RGO with hydrothermal method was very sensitive to CO [19]. Liu et al. prepared a flower-like morphology of ZnO in order to provide more gas adsorption sites by increasing the specific surface area and designed a flower-like ZnO with RGO composite structure, which could detect ultra-low concentrations of nitrogen dioxide gas [20]. However, most of the sensors mentioned above must have good gas sensing properties in the temperature range of 200–300 °C. Although a few sensors could detect target gas at room temperature, the gas sensing properties were not good enough and the synthetic processes were relatively complex.

Cobalt carbonate hydroxide hydrate (Co(CO_3_)_0.5_OH•0.11H_2_O), referred as CCHH has been usually utilized as the precursor to obtain Co_3_O_4_ by annealing process at 300–500 °C [21,22,23]. CCHH has the potential to become a promising gas sensing material due to its simple synthesis process, high specific surface area and large three-dimensional space. However, to the best of our knowledge, no work has been reported on the use of CCHH as an efficient gas sensing material. The reason CCHH has not used as a sensing material before is the low conductivity of CCHH as the intrinsic limitation for its gas sensing application. Therefore, improving its conductivity is extremely critical for CCHH used in the gas sensing field. A favorable way to overcome this problem is to incorporate CCHH with high conductivity materials in a proper manner, which can efficiently facilitate the electron transport and improve the gas sensing properties.

This paper proposed a novel hybrid structure sensor based on CCHH and RGO, designed for room temperature NH_3_ detection. The hybrid structure consisted of CCHH and RGO, in which CCHH was designed as a gas sensing material and RGO as a conductive material. We expected to achieve the improvement of SNR and the sensitivity by obtaining the optimal proportion of RGO, because the proportion of RGO in the hybrid structure was directly related to the sensitivity.

## 2. Materials and Methods

### 2.1. Preparation of CCHH with RGO

GO (20 mg, #XF002-1, XF Nano Inc., Nanjing, China) was dispersed in deionized water (60 mL), vibrated and sonicated for 4 h. After GO dispersed evenly in the solution, cobalt chloride hexahydrate (20 mmol, #20160823, Guangdong Guanghua Sic-Tech Co., Ltd., Shantou, China) and urea (20 mmol, Tianli Chemical Reagent Co. Ltd., Tianjin, China) were added into the dispersed solution, stirred and sonicated for 3 min. When the two chemicals were completely dissolved and evenly distributed in the solution, the solution was transferred into a reaction kettle to conduct hydrothermal reactions for 3 h under 130 °C. The solution fabricated by hydrothermal reaction was centrifuged and washed with deionized water several times. The obtained material was dried in oven under 60 °C for 12 h and the final product was obtained. The fabrication process of CCHH with RGO hybrid structure was shown in Figure 1. For comparative experiments, GO was processed in the same experimental procedure without cobalt chloride hexahydrate and urea. The results of the two experiments were attentively compared.

### 2.2. Fabrication of CCHH-RGO Gas Sensor

The previously cleaned target solution mainly being CCHH with RGO hybrid material, was transferred onto the fresh interdigital electrode (IDE) and heated in oven under 50 °C for 10 min and then the gas sensor named as CCHH-RGO was obtained.

In order to obtain the highest sensing response, the effect of the proportion of RGO in the hybrid structure on gas sensing response has been studied. According to the different RGO proportion, several gas sensors with different RGO mass proportion were fabricated at the same time and listed in Table 1.

### 2.3. Material Characterization

The structure and morphology of CCHH-RGO were investigated with scanning electron microscopy (SEM, S-4800, Hitachi, Tokyo, Japan). X-ray photoelectron spectroscopy (XPS, Axis Ultra DLD, Kratos Inc., Manchester, UK) and X-ray diffraction (XRD, XRD-6100, Shimadzu Corporation, Kyoto, Japan) were used to characterize the elemental composition and state of the composites, respectively.

### 2.4. Gas Sensing Detection

The whole ammonia gas sensing system was based on the homemade testing chamber shown in Figure 2 and the total volume of the system was calculated as 4.67 L. Desiccant in the flask was sodium hydroxide used to convert ammonia solution into dry ammonia gas, which was passed into the testing chamber by an air pump (MEDOVP0125-V1005-P2-1411, Nitto Kohki, Tokyo, Japan). The calculation details of dry ammonia gas concentration were included in the Appendix A. The test results were recorded by the multimeter (2000, Keithley Instruments, Cleveland, OH, USA). The entire test experiment was carried out at the temperature of 25 ± 2 °C and the humidity of 30 ± 5%.

## 3. Results and Discussion

### 3.1. Characterization of Sensing Materials

The structure and morphology of the synthesized CCHH-RGO was characterized by various instruments. Considering the influence of different RGO proportion on the structure and gas sensing properties, CCHH-RGO-0.4 with the optimized gas sensitivity was systematically characterized as a typical one.

Figure 3a illustrated a SEM image of CCHH-RGO-0.4, revealing the 3-dimensional (3D) CCHH connected to each other by RGO. The enlarged image showed a sisal-like structure in Figure 3b, which could provide airflow pathway and large adsorption surface area of gas molecules. Therefore, CCHH was a very suitable candidate for gas adsorption material in terms of its 3D structure. Although the proportion of RGO was very low, it could be clearly seen that RGO shown in the white square in Figure 3a was uniformly distributed on the surface of CCHH. In such a structure, uniformly dispersed RGO helped CCHH achieve rapid charge transfer. To investigate the composition of the obtained sample, the XRD pattern of CCHH-RGO-0.4 was shown in Figure 3c. All of these characteristic peaks in the pattern were well indexed to cobalt carbonate hydroxide hydrate (JCPDS No. 48-0084). No peaks of other impurities such as Co_3_O_4_ were detected in the sample pattern, meaning that high purity CCHH was well synthesized in the RGO dispersed solution with the hydrothermal method.

For further characterization, we performed XPS analysis of CCHH-RGO-0.4 to further investigate the chemical composition of the sample. The Co 2p spectrum displayed two major peaks with the Binding energy (BE) values at 781.2 and 797.1 eV in the Figure 4a, which was well matched with the Co 2p3/2 and Co 2p1/2 peaks, respectively [24,25]. Meanwhile, two sub-peaks observed beside the main peaks at 785.8 and 803 eV in the spectrum clearly indicated the existence of Co^2+^ in the sample. Data from XPS and XRD showed that CCHH-RGO-0.4 was mainly composed of Co^2+^. The existence of sub-peaks in the spectra further revealed that Co^2+^ was not oxidized to the Co_3_O_4_, which well matched the research results by Varghese [26]. The C 1s peak of the sample at 284.9 eV in the Figure 4b was divided into a group of peaks, which mainly included C-C, C-OH, C-O-C and CO_3_^2−^ at 284.9, 286.5, 287.1 and 289.7 eV, separately. Combined with the C 1s peaks of the sample and GO, we concluded that GO was completely reduced to RGO due to the drastic reduction of oxygen-containing functional groups such as C-OH and C-O-C. Meanwhile, the new peak appearing at 289.7 represented presence of CO_3_^2−^, which was mainly derived from the formation of CCHH.

### 3.2. Sensing Performance

In order to study the gas sensing properties of CCHH and RGO hybrid sensors, a series of gas sensing tests were implemented in the proposed self-made test system at six different concentrations of ammonia at room temperature. The resistance values of several sensors changed with the injection and discharge of ammonia. The gas sensing response can be obtained as follow equation.
$$\mathrm{Response}\left(\%\right)=\frac{R_{\mathrm{ammonia}}-R_{\mathrm{air}}}{R_{\mathrm{air}}}\times 100\%$$

R_ammonia_ and R_air_ are the electrical resistance values with and without ammonia gas, respectively. The gas sensing response is defined as a ratio expressed in percentage.

As the ammonia gas in, the response curves of sensors began to increase rapidly and then slowly stabilized at a fixed maximum. Next, the response values decreased rapidly and returned to near the initial value after ammonia gas off. The black, red, blue and green curves correspond to the responses of four sensors (named RGO, CCHH-RGO-16, CCHH-RGO-4 and CCHH-RGO-0.4) to six concentrations of ammonia gas (1, 2.5, 5, 10, 25, 50 ppm) in Figure 5. In addition to these four sensors, the gas sensing responses of the remaining two sensors named CCHH and CCHH-RGO-0.1 were not given in the manuscript but provided in as shown in Appendix A. The base resistance values of these two sensors (more than 100 MΩ) leaded to the signal current entering nanoampere level, bringing two fatal flaws. One of them, such a low signal current would cause the influence of background noise to increase dramatically and greatly interfere with the test results. Another, this level of signal current requires ultra-high precision in the test equipment, which meant a significant increase in signal processing circuit costs.

Figure 5 illustrated the gas sensing responses of four sensors at different ammonia concentrations. The gas sensing responses of these sensors gradually increased with the ammonia concentration and tended to be saturated and did not continue to increase when the ammonia concentration reached a higher value. As it could be seen clearly from the response curves, the response of RGO exhibited the lowest value, CCHH-RGO-4 exhibited higher value than CCHH-RGO-16, while CCHH-RGO-0.4 showed the highest response compared to the other three sensors. From the test results, the gas sensing response of the sensor to ammonia gas increased with the decrease of RGO mass proportion in the hybrid structure. When the RGO mass proportion reached 0.4%, the gas sensing response was highest. As the RGO proportion continued to decrease, it would be very difficult to conduct gas test due to the great resistivity of the sensor such as CCHH and CCHH-RGO-0.1.

To study the gas sensing mechanism of the hybrid structure sensor, we performed a gas sensing test on the 10 ppm ammonia gas for four sensors with a gas on (360 s)/gas off (1200 s) cycle of 1560 s at the same time. Figure 6a showed the gas sensing curves of four sensors to 10 ppm ammonia. The gas sensing response, response time and recovery time of each sensor were shown in Figure 6b. RGO showed the lowest response (3.5%), the longest response time (272 s) and the longest recovery time (895 s). The gas sensing response increased, the response time and the recovery time decreased with the decrease of RGO mass proportion in the hybrid structure. When the RGO mass proportion reaches 0.4%, CCHH-RGO-0.4 showed the highest response (43.7%), the least response time (184 s) and the least recovery time (575 s) toward 10 ppm ammonia.

Similar to metal oxide, there were a large number of lattice defects, surface discontinuities and a certain concentration of electron donor and acceptor energy levels on the surface of CCHH, which could exchange electrons with oxygen molecules in the air to form oxygen anions ($O_{2}^{-}$) at room temperature and conduct charge on the surface with the formation of space charge layer or space depletion region [27].

When ammonia gas reacted with the oxygen anion on the surface, ammonia molecules would release an electron to form ammonium ions. While the electron could be smoothly transferred to the surface of CCHH and then further diverted to RGO due to coupling effect between CCHH and RGO, which could give rise to the resistivity increase of the sensor owing to the prepared RGO being p-type material. Once the ammonia gas was removed, the ammonium ions would be converted to ammonia molecules and the resistivity of the sensor would decrease. Besides, CCHH prepared in this experiment was a sisal-like structure with a high specific surface area and the structure was conducive to the gas sensing response increase because the sisal-like structure was suitable for gas circulation.

The introduction of RGO greatly reduced the resistivity of the sensor due to the high conductivity of RGO, which not only facilitated the electronic conduction of the sensor after adsorbing gases but also increased the signal current to improve SNR and weaken the effect of noise. CCHH covered with RGO could sense tiny charge changes from adsorbed gases and provide a basis for realizing room temperature gas sensing tests. However, the proportion of RGO was not the more the better. The experimental results showed that with the increase of RGO proportion, the response of the sensor decreased gradually. The main reason may be that excessive RGO covered the surface of CCHH, which prevented the gas molecules from reacting with the CCHH, thus preventing the change of conductivity.

The exposure of the CCHH-RGO-0.4 to four continuous cycles of 10 ppm ammonia was shown in Figure 7a. The sensor showed approximately the same response value and the response/recovery time, revealing its excellent repeatability under room temperature. In order to test the selectivity of the sensor to ammonia gas (10 ppm), four other kinds of 100 ppm gases were tested under exactly the same test conditions, including acetone, isopropanol, ethanol and formaldehyde shown in Figure 7b. The device responded slightly to acetone and formaldehyde but hardly to isopropanol and ethanol. However, in sharp contrast, the device showed a very high response to ammonia, suggesting the strong selectivity of CCHH-RGO-0.4 towards ammonia.

There were several performance comparisons of different sensors reported by other journals. The sensing material, operating temperature, gas concentration and response of those sensors were listed in Table 2. Ag/ZnO and Pt/SnO_2_ as the sensing materials, could obtain a good gas sensing response at a high operating temperature. Although the remaining four sensing materials could be used for gas sensors at room temperature, their gas sensing responses were relatively low. Compared with these sensors, the proposed sensor in this paper with the simple fabrication process, not only could achieve gas sensing test at room temperature but also had excellent gas sensing performances.

## 4. Conclusions

In summary, the paper propose a novel CCHH-RGO gas sensor fabricated by a by one-step hydrothermal method. The simple synthesis not only simplified the experimental process but also facilitates mass production. The prepared sisal-like CCHH with high specific surface area could provide abundant gas adsorption interface and a suitable pathway for gas molecules. Meanwhile, the synthesized RGO uniformly dispersed on CCHH surface significantly improved the signal current and gas sensitivity caused by gas molecular adsorption. When the mass proportion of RGO was 0.4%, CCHH-RGO hybrid structure had the optimized gas sensing response, reach to 9% to 1 ppm ammonia. In addition, it could be found that CCHH-RGO hybrid structure had a good selectivity to ammonia by comparing several different gases as acetone, isopropanol, ethanol and formaldehyde. We believe that the novel hybrid sensor has the potential to be used in mass-produced gas sensors and is valuable for the future research on gas sensors.

## Figures and Tables

**Figure 1 sensors-19-00615-f001:**
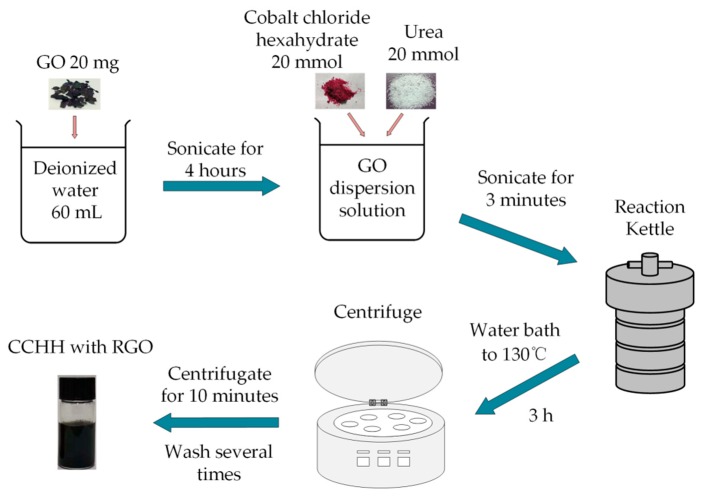
Fabrication flowchart of cobalt carbonate hydroxide hydrate (CCHH) with RGO hybrid structure.

**Figure 2 sensors-19-00615-f002:**
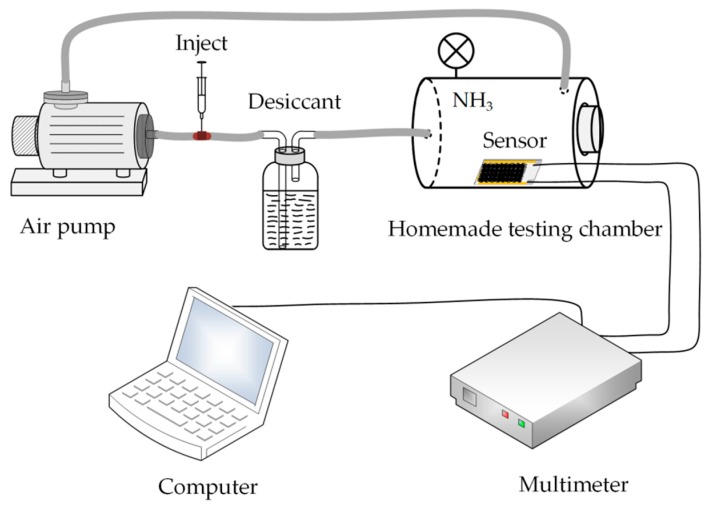
Schematic diagram of the gas detection system.

**Figure 3 sensors-19-00615-f003:**
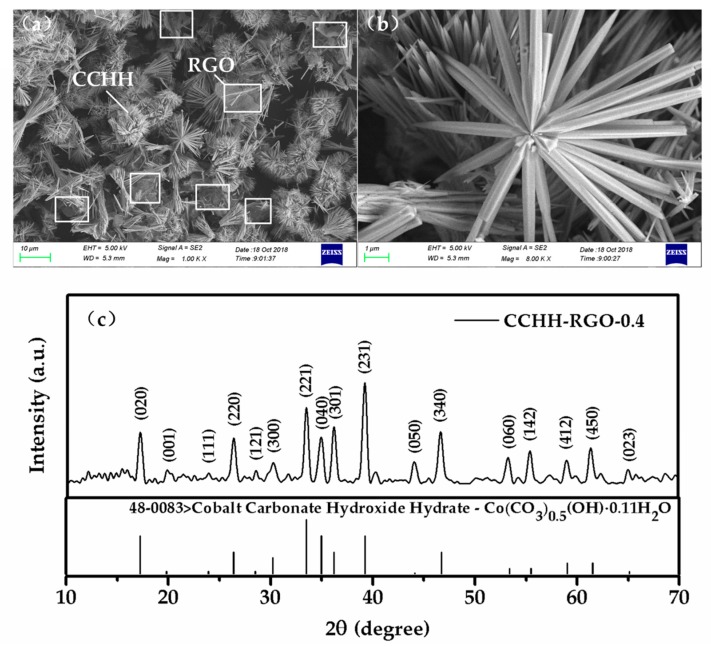
(**a**) Scanning electron microscope (SEM) image of CCHH-RGO-0.4; (**b**) the enlarged SEM image of CCHH-RGO-0.4; (**c**) X-ray diffraction (XRD) pattern of CCHH-RGO-0.4.

**Figure 4 sensors-19-00615-f004:**
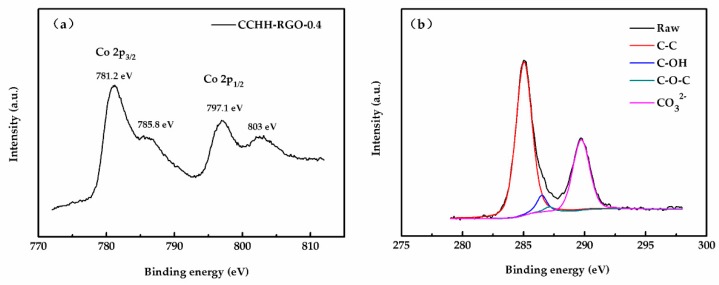
(**a**) XPS spectrum of the Co 2p composite; (**b**) XPS spectrum of the C 1s composite.

**Figure 5 sensors-19-00615-f005:**
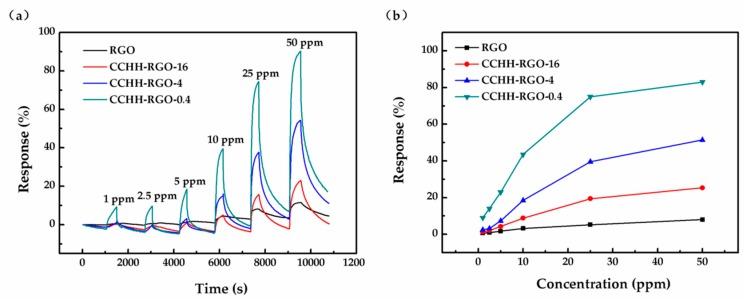
(**a**) Gas sensing response of four sensors under six different concentrations of ammonia; (**b**) Response with different concentrations of ammonia ranging from 1 to 50 ppm.

**Figure 6 sensors-19-00615-f006:**
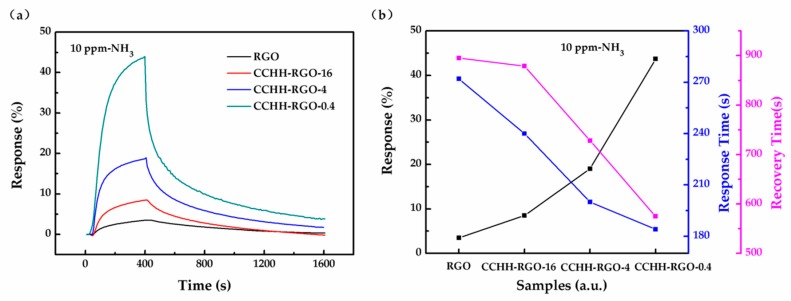
(**a**) Gas sensing response curve of four sensors to 10 ppm ammonia; (**b**) Gas sensing response, response time and recovery time of four sensors for 10 ppm ammonia.

**Figure 7 sensors-19-00615-f007:**
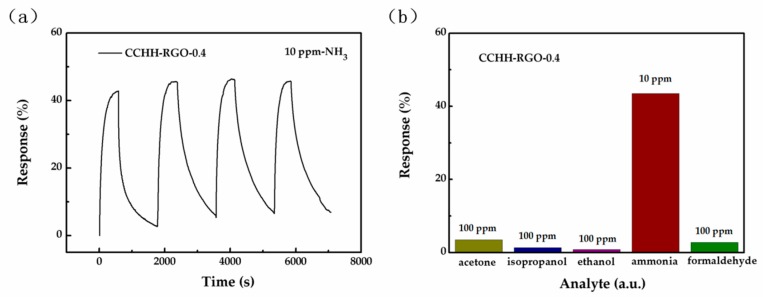
(**a**) Gas sensing response of the gas sensor after four continuous cycles under 10 ppm ammonia; (**b**) Selectivity response when exposed to acetone, isopropanol, ethanol, ammonia and formaldehyde.

**Table 1 sensors-19-00615-t001:** Relationship of sensor name and sensing material.

Sensors	CCHH	CCHH-RGO-0.1	CCHH-RGO-0.4	CCHH-RGO-4	CCHH-RGO-16	RGO
Mass proportion of RGO/CCHH (wt %)	0	0.1	0.4	4	16	100

**Table 2 sensors-19-00615-t002:** Comparison of various indicators between different ammonia sensors.

Materials	Temperature (°C)	Concentration (ppm)	Response (%)	Reference
Ag/ZnO	150	10	29.5	[28]
Pt/SnO_2_	115	50	25	[29]
RGO/Graphene	25	0.5	2.88	[30]
PANI/SnO_2_	25	10	5	[31]
Modified-CNT	25	1.5	0.65	[32]
Graphene/TiO_2_	25	5	1.25	[33]
CCHH/RGO	25	1(10)	9(43)	This Work

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
