# Peer review of "Simple Synthesis of Cobalt Carbonate Hydroxide Hydrate and Reduced Graphene Oxide Hybrid Structure for High-Performance Room Temperature NH3 Sensor"

_sensors, 2019, doi:10.3390/s19030615_

Reviewer 1 Report

Text present description of preparation and investigation of NH3 sensor
based on cobalt carbonate hydroxide hydrate and reduced graphene oxide.
The text is concise and easy to read. Still, some parts of it can be
improved.

1) The text is written in a generally understandable language, but in
some places language corrections are required, eg. line 26 'but also is
easy to achieve mass 26 production due to the good stability and simple
synthesis process' or line 208 'However, the proportion of RGO is not
the more the better.' where something is probably wrong.

2) I do not agree that 'The introduction of RGO greatly reduces the
resistivity of the sensor due to the high conductivity of RGO, ..., but
also increases the signal current to suppress background noise.' (line
204-206). The is no noise suppression. The useful signal is larger,
hence the signal to noise ratio is better. Keithley 2000 is not designed
to measure high resistances. The 'noise' that the authors mention is the
effect of the limitations of the measuring device. Similarly, the
statement 'We expect to achieve the suppression of noise interference'
(line 78) is imprecise, misleading and should be removed. Line 22, 'The
hybrid structure can suppress noise interfer' has the same problem.

3) Authors investigate sensor in dry gas. In real life, some humidity is
always present. I would like the authors to comment on how the presence
of humidity can affect the properties of the sensor.

Author Response

Dear Prof. Editor:

Thank you very much for giving us an opportunity to revise our
manuscript, we appreciate very much the editor and reviewers for their
kind and constructive comments and suggestions on our manuscript
entitled "Simple Synthesis of Cobalt Carbonate Hydroxide Hydrate and RGO
Hybrid Structure for High-Performance Room Temperature NH3 Sensor"
(Manuscript ID: sensors-422175). We have studied reviewer’s comments
carefully and have made revision which marked in red in the paper. We
have tried our best to revise our manuscript according to the comments.
Please find the revised version, which we would like to submit for your
kind consideration.

We would like to express our great appreciation to you and reviewers for
comments on our paper. Looking forward to hearing from you.

Thank you and best regards.

Yours sincerely,

Prof. Xin Li

Corresponding author (E-mail: lx@mail.xjtu.edu.cn)

-------------------------------------------------------------------------------------------------------

The following are point-to-point response to the reviewers’ comments. 

Reviewer 1

Text present description of preparation and investigation of NH3 sensor
based on cobalt carbonate hydroxide hydrate and reduced graphene oxide.
The text is concise and easy to read. Still, some parts of it can be
improved.

1. The text is written in a generally understandable language, but in
some places language corrections are required, eg. line 26 'but also is
easy to achieve mass 26 production due to the good stability and simple
synthesis process' or line 208 'However, the proportion of RGO is not
the more the better.' where something is probably wrong.

Answer: We thank and appreciate the reviewer’s comment. There were some
language errors and inconsistent descriptions (such as between line 26
and line 208) in the manuscript. We have corrected them and marked in
red in the revised version.

2. I do not agree that 'The introduction of RGO greatly reduces the
resistivity of the sensor due to the high conductivity of RGO, ..., but
also increases the signal current to suppress background noise.' (line
204-206). There is no noise suppression. The useful signal is larger,
hence the signal to noise ratio is better. Keithley 2000 is not designed
to measure high resistances. The 'noise' that the authors mention is the
effect of the limitations of the measuring device. Similarly, the
statement 'We expect to achieve the suppression of noise interference'
(line 78) is imprecise, misleading and should be removed. Line 22, 'The
hybrid structure can suppress noise interference' has the same problem.

Answer: We thank and appreciate the reviewer’s comment. The reviewer's
view is correct that the previous statements on suppressing noise were
unreasonable in the manuscript. The introduction of RGO could reduce the
resistivity of the sensor and increase the signal current and the
signal-to-noise ratio, which just weakened the effect of noise, rather
than reduced the noise. We have reinterpreted the effect of the hybrid
structure on noise in the corresponding region of the manuscript.

3. Authors investigate sensor in dry gas. In real life, some humidity is
always present. I would like the authors to comment on how the presence
of humidity can affect the properties of the sensor.

Answer: Thank the reviewer for the comment. We have supplemented the
test results of the sensor's gas sensing response to 10 ppm ammonia at
different relative humidity. Figure shows the gas sensing response of
CCHH-RGO-0.4 to 10 ppm ammonia under four relative humidity (including
10%, 30%, 50% and 75%). When relative humidity is low, the gas sensing
response is relatively poor. However, the gas sensing response increases
gradually with the increase of relative humidity. It can be concluded
that moisture has a very good promoting effect on gas sensing response.

Reviewer 2 Report

The submitted manuscript presents results on the performance of a
CCHH-RGO composite as an ammonia sensor at room temperature. The
manuscript is well organized, however authors need to carefully revise
it since typographical and wording errors were found. Some comments and
suggestions are included in the attached pdf file. Some other comments
are given below as to improve the quality of the paper. In this sense, a
major revision is recommended.

Use past tense when writing a report (sometimes present tense was used
in the manuscript).

Do the error/uncertainty propagation so that all numerical results have
errors included; include error bars in graphs.

In the Introduction section, it would be good to mention typical
composition of exhaust gases, related to the application of the proposed
sensor.

Text on all figures would be better to have the same size.

Unfortunately I could not have access to the supplementary material
("SI") mentioned in the manuscript. Results of 0.1 % RGO sample are
supposed to be shown there and I wonder whether it was difficult to do a
proper weighting of the materials to differentiate 0.1 and 0.4 % samples.

In page 5, it was stated that GO was completely reduced in sample with a
0.4 % RGO. Were the results similar for the higher RGO content samples?

Regarding the RGO coverage/superposition of the CCHH surface (last
paragraph of page 6), it would be better to include SEM images of a
sample with a higher RGO content than 0.4 %.

There is no mention in the manuscript regarding reproducibility of the
sensing performance for different sensors (I mean for sensors prepared
at different batches with exactly the same nominal composition).

Are you considering doing some life time experiments? How different is
the life time for sensors included in table 2?

In table 2, response value of your sensor for 1 ppm NH3 was included.
Since some other authors have results for 10 ppm as well, you could also
include the response value for 10 ppm NH3.

Author Response

Dear Prof. Editor:

Thank you very much for giving us an opportunity to revise our
manuscript, we appreciate very much the editor and reviewers for their
kind and constructive comments and suggestions on our manuscript
entitled "Simple Synthesis of Cobalt Carbonate Hydroxide Hydrate and RGO
Hybrid Structure for High-Performance Room Temperature NH3 Sensor"
(Manuscript ID: sensors-422175). We have studied reviewer’s comments
carefully and have made revision which marked in red in the paper. We
have tried our best to revise our manuscript according to the comments.
Please find the revised version, which we would like to submit for your
kind consideration.

We would like to express our great appreciation to you and reviewers for
comments on our paper. Looking forward to hearing from you.

Thank you and best regards.

Yours sincerely,

Prof. Xin Li

Corresponding author (E-mail: lx@mail.xjtu.edu.cn)

-------------------------------------------------------------------------------------------------------

The following are point-to-point response to the reviewers’ comments. 

Reviewer 2

The submitted manuscript presents results on the performance of a
CCHH-RGO composite as an ammonia sensor at room temperature. The
manuscript is well organized, however authors need to carefully revise
it since typographical and wording errors were found. Some comments and
suggestions are included in the attached pdf file. Some other comments
are given below as to improve the quality of the paper. In this sense, a
major revision is recommended.

1. Use past tense when writing a report (sometimes present tense was
used in the manuscript). Do the error/uncertainty propagation so that
all numerical results have errors included; include error bars in
graphs. In the Introduction section, it would be good to mention typical
composition of exhaust gases, related to the application of the proposed
sensor. Text on all figures would be better to have the same size.

Answer: We are very grateful to the reviewer for his/her careful review
and kind reminder. There were some grammatical errors in the manuscript.
We have corrected them and marked in red in the revised version. As for
the figures in the manuscript, we have revised them according to the
comments of the reviewer.

2. Unfortunately I could not have access to the supplementary material
("SI") mentioned in the manuscript. Results of 0.1 % RGO sample are
supposed to be shown there and I wonder whether it was difficult to do a
proper weighting of the materials to differentiate 0.1 and 0.4 % samples.

Answer: Thank the reviewer for the comment. The key is not the
difficulty of weighing of the materials to differentiate 0.1 and 0.4 %
samples, but the problem of gas sensing test of 0.1 % RGO sample.
Because of the low conductivity of CCHH and CCHH-RGO-0.1 (much lower
than the conductivity of the other four samples named RGO, CCHH-RGO-0.4,
CCHH-RGO-4, CCHH-RGO-16), the test current is very small (reaching the
nanoampere level), exceeding the accuracy of the test equipment
(multimeter, 2000, Keithley Instruments). Despite replacing the original
test equipment with Agilent 4155c Semiconductor Parameter Analyzer,
ultra-low test current still leads to a bad effect of the environmental
noise on test results shown in the figure. Under these circumstances, we
consider that it would be more appropriate to display the data of these
two samples in SI.

3. In page 5, it was stated that GO was completely reduced in sample
with a 0.4 % RGO. Were the results similar for the higher RGO content
samples? Regarding the RGO coverage/superposition of the CCHH surface
(last paragraph of page 6), it would be better to include SEM images of
a sample with a higher RGO content than 0.4 %.

Answer: Thank the reviewer for the comment. GO can be well reduced by
the hydrothermal method. We used the same preparation process (the same
preparation temperature, the same preparation time, but the different
amount of RGO), GO in each sample is completely reduced. According to
the requirements of the reviewer, we provide a SEM image of a sample
with a higher RGO content than 0.4 %. As shown in the following figure.

4. There is no mention in the manuscript regarding reproducibility of
the sensing performance for different sensors (I mean for sensors
prepared at different batches with exactly the same nominal
composition). Are you considering doing some life time experiments? How
different is the life time for sensors included in table 2?

Answer: Thank the reviewer for the kind reminder. We have conducted
experiments in the sensing performance for sensors prepared at different
batches with exactly the same nominal composition according to the
requirement of the reviewer. The gas sensing response of three sensors
of different batches with the same nominal composition to three
continuous cycles under 10ppm ammonia is shown in the following figure.
As can be seen in the figure, the gas sensing responses of sensors in
different batches are similar. It can be seen that the hybrid structure
sensor is relatively stable and reliable. For the lifetime of the
sensor, we have also conducted a series of tests. Due to the limitation
of the revision time of the manuscript, it is impossible for us to test
the lifetime of the sensor for a long time. At present, we can only
provide short-term test results as shown in the figure. Next, we will
continue to test the lifetime of the sensor in the future work.

5. In table 2, response value of your sensor for 1 ppm NH3 was included.
Since some other authors have results for 10 ppm as well, you could also
include the response value for 10 ppm NH3.

Answer: Thank the reviewer for the comment. We have supplemented the
experimental results of the proposed sensor for 10 ppm NH3 in table 2 in
the revised manuscript.

Reviewer 3 Report

The authors present a range of ammonia sensors based on the variation of
the resistance of the hybrid sensitive material. The hybrid structure
consists of cobalt carbonate hydroxide hydrate (CCHH) and reduced
graphene oxide (RGO). CCHH acts as the gas sensing material and RGO as
the conductive material that increases the conductivity of CCHH. The
effect of the proportion of RGO in the hybrid material has been also
studied by fabricating and testing a range of sensors with different
amounts of RGO. The sensors show a good response at room temperature,
especially the sensor with 0.4% of RGO.

However, I have some questions that the authors should address before
publication:

Section 2.1: The precursor materials have been synthesised by the
authors? If yes, specify the method used to do this. If not, specify the
company they were purchased from and the product identifier.

Section 2.1: Detail the quantities of RGO added to obtain the different
proportions of RGO in the hybrid materials.

Section 2.2: Explain how the transfer of the material from the solution
to the interdigital electrode is performed, specifying the amount of
material and the method of transfer.

Section 3.2: The response of sensors CCCH and CCCH-RGO 0.1 should be
included in this section instead of including them as supplementary
material.

Section 3.2: How have you calculated the response time? Rise time from
10 to 90%? From 0 to 100%?

Section 3.2: Specify the recovery times of your sensors.

Figure 7a, caption: Profile of four gas sensors? I would say it is the
response of only one sensor to four cycles.

Section 3.2: Specify the sensitivity and resolution of your sensors and
compare these values with other ammonia sensors based on resistance
measurements.

This work needs a deep revision of English.

Sensors is a journal that typically pays attention to the presentation
and the graphical aspect of the articles. In this sense, all the figures
and specifically figures 1 and 2 should be improved and/or re-designed.

Consequently, I consider the content of this study suitable for
publication in Sensors after a minor but deep revision.

Author Response

Dear Prof. Editor:

Thank you very much for giving us an opportunity to revise our
manuscript, we appreciate very much the editor and reviewers for their
kind and constructive comments and suggestions on our manuscript
entitled "Simple Synthesis of Cobalt Carbonate Hydroxide Hydrate and RGO
Hybrid Structure for High-Performance Room Temperature NH3 Sensor"
(Manuscript ID: sensors-422175). We have studied reviewer’s comments
carefully and have made revision which marked in red in the paper. We
have tried our best to revise our manuscript according to the comments.
Please find the revised version, which we would like to submit for your
kind consideration.

We would like to express our great appreciation to you and reviewers for
comments on our paper. Looking forward to hearing from you.

Thank you and best regards.

Yours sincerely,

Prof. Xin Li

Corresponding author (E-mail: lx@mail.xjtu.edu.cn)

-------------------------------------------------------------------------------------------------------

The following are point-to-point response to the reviewers’ comments. 

Reviewer 3

The authors present a range of ammonia sensors based on the variation of
the resistance of the hybrid sensitive material. The hybrid structure
consists of cobalt carbonate hydroxide hydrate (CCHH) and reduced
graphene oxide (RGO). CCHH acts as the gas sensing material and RGO as
the conductive material that increases the conductivity of CCHH. The
effect of the proportion of RGO in the hybrid material has been also
studied by fabricating and testing a range of sensors with different
amounts of RGO. The sensors show a good response at room temperature,
especially the sensor with 0.4% of RGO. However, I have some questions
that the authors should address before publication:

1. Section 2.1: The precursor materials have been synthesized by the
authors? If yes, specify the method used to do this. If not, specify the
company they were purchased from and the product identifier.

Answer: We thank and appreciate the reviewer’s comment. The precursor
materials in the synthesis process all were purchased online. Detailed
information on the companies and the product identifiers have been
provided in the revised manuscript.

2. Detail the quantities of RGO added to obtain the different
proportions of RGO in the hybrid materials.

Answer: Thank the reviewer for the kind reminder. We prepared six hybrid
structural materials with different proportions of RGO. The compositions
and quantities of each hybrid structural material are shown in the table
below.

Table 1

Sensors

CCHH

CCHH-

RGO-0.1

CCHH-

RGO-0.4

CCHH-RGO-4

CCHH-RGO-16

RGO

Mass proportion of RGO/CCHH   (wt %)

0

0.1

0.4

4

16

100

The mass   of RGO (mg)

0

4

20

100

200

200

The mass   of CCHH (mg)

4760

4760

4760

2380

1190

0

The amount of material on IDE (mL)

0.2

0.2

0.2

0.4

0.8

0.8

3. Explain how the transfer of the material from the solution to the
interdigital electrode is performed, specifying the amount of material
and the method of transfer.

Answer: Thank the reviewer for the comment. The material was synthesized
by hydrothermal method, which was eventually prepared into 10 ml
dispersion solution of the material after cleaned. 0.2-0.8 ml of
dispersion solution (as shown in the above Table 1) was sucked with a
tube and dropped onto the interdigital electrode.

4. The response of sensors CCCH and CCCH-RGO 0.1 should be included in
this section instead of including them as supplementary material.

Answer: We thank and appreciate the reviewer’s comment. Because of the
low conductivity of CCHH and CCHH-RGO-0.1 (much lower than the
conductivity of the other four samples named RGO, CCHH-RGO-0.4,
CCHH-RGO-4, CCHH-RGO-16), the test current is very small (reaching the
nanoampere level), exceeding the accuracy of the test equipment
(multimeter, 2000, Keithley Instruments). Despite replacing the original
test equipment with Agilent 4155c Semiconductor Parameter Analyzer,
ultra-low test current still leads to a bad effect of the environmental
noise on test results shown in the figure. Under these circumstances, we
consider that it would be more appropriate to display the data of these
two samples in SI.

5. How have you calculated the response time? Rise time from 10 to 90%?
From 0 to 100%? Specify the recovery times of your sensors.

Answer: Thank the reviewer for the comment. Rising time and recovery
time are defined as the time from 0 to 90% of the response and the time
from 100 to 10% of the response, respectively. We have supplemented the
recovery time data in the revised manuscript.

6. Figure 7a, caption: Profile of four gas sensors? I would say it is
the response of only one sensor to four cycles.

Answer: We thank and appreciate the reviewer’s comment. There was a
mistake in the caption in the manuscript. We have corrected it and
marked in red in the revised version.

7. Section 3.2: Specify the sensitivity and resolution of your sensors
and compare these values with other ammonia sensors based on resistance
measurements.

Answer: Thank the reviewer for the comment. Since the sensor with the
hybrid structure is easily saturated in high concentration ammonia, we
can only calculate the sensitivity of the sensor to ammonia at low
concentration (i.e. linear region). The sensitivity of all sensors is
shown in the table below.

Sensors

CCHH-

RGO-0.4

CCHH-

RGO-4

CCHH-

RGO-16

RGO

The   sensitivity of the sensor ( %/ppm)

3.83

2.04

0.83

0.29

8. This work needs a deep revision of English. Sensors is a journal that
typically pays attention to the presentation and the graphical aspect of
the articles. In this sense, all the figures and specifically figures 1
and 2 should be improved and/or re-designed.

Answer: We thank and appreciate the reviewer's comment. We have improved
our English expression in the revised manuscript. The figures mentioned
by the reviewer have been greatly revised according to the comments of
the reviewer.
